# Modelling atomic and nanoscale structure in the silicon–oxygen system through active machine learning

Linus C. Erhard [1], Jochen Rohrer[1] ✉, Karsten Albe [1] ✉ &
Volker L. Deringer [2] ✉

Silicon–oxygen compounds are among the most important ones in the natural sciences, occurring as building blocks in minerals and being used in semiconductors and catalysis. Beyond the well-known silicon dioxide, there are phases with different stoichiometric composition and nanostructured composites. One of the key challenges in understanding the Si–O system is therefore to accurately account for its nanoscale heterogeneity beyond the length scale of individual atoms. Here we show that a unified computational description of the full Si–O system is indeed possible, based on atomistic machine learning coupled to an active-learning workflow. We showcase applications to very-high-pressure silica, to surfaces and aerogels, and to the structure of amorphous silicon monoxide. In a wider context, our work illustrates how structural complexity in functional materials beyond the atomic and few-nanometre length scales can be captured with active machine learning.

Elemental silicon and its oxide, silica ($SiO_2$), are widely studied building blocks of the world around us[1]: from minerals in geology to silicon-based computing architectures; thin-film solar cells in which amorphous silicon is the active material[2]; or zeolite catalysts based on the $SiO_2$ parent composition[3]. Some of these materials have a single phase and are precisely defined on the atomic scale, whereas others show longer-ranging, hierarchical structures and varying degrees of disorder. For example, silica aerogels contain pores with sizes of 5–100 nm, leading to very low thermal conductivity and making aerogels promising candidates for thermal insulation[4]. Under pressure, $SiO_2$ shows amorphous–amorphous transitions to structures exceeding sixfold coordination[5], crystallisation from the amorphous phase under shock compression[6], and conversely the formation of complex disordered phases from crystalline $SiO_2$[7]. Beyond fundamental studies, there is much technological importance in silicon–oxygen phases with nanoscale structure—the interface between Si and $SiO_2$ is essential in silicon metal-oxide semiconductors, and defects at this interface have been investigated for decades[8–10].

A material in the binary silicon–oxygen system which is in fact dominated by such interfaces is the so-called silicon monoxide (SiO). The structure of SiO was controversially discussed for long[11,12]; today, it is known as a nanoscopic mixture of amorphous Si and $SiO_2$[13–15]. Initial applications of SiO have been in protective layers for mirrors[16] or dielectrics for thin-film capacitors[17]; more recently, the same material has emerged as a promising anode material for lithium-ion batteries[18,19]. However, to be able to fully exploit SiO in next-generation energy-storage solutions, it would be valuable to understand the features of the nanoscopic structure on an atomistic level.

To develop atomic-scale models of complex materials such as SiO, molecular-dynamics (MD) computer simulations have become a central research tool. While there are now plenty of interatomic potentials for silicon[20–22] and silica[23–25], the number of potentials for the mixed (i.e., full binary) system is limited due to its chemical complexity[26–29]. Alongside established, empirically fitted potentials based on physical models, alternatives based on large datasets and machine learning (ML) have emerged in recent years. These models have been fitted for

[1]Institute of Materials Science, Technische Universität Darmstadt, Otto-Berndt-Strasse 3, D-64287 Darmstadt, Germany. [2]Department of Chemistry, Inorganic Chemistry Laboratory, University of Oxford, Oxford OX1 3QR, United Kingdom. ✉e-mail: rohrer@mm.tu-darmstadt.de; albe@mm.tu-darmstadt.de; volker.deringer@chem.ox.ac.uk

silicon[30] as well as silica[31] and also for the more complex silica–water system[32]. ML potentials promise the accuracy of first-principles methods such as density-functional theory (DFT) for a small fraction of the cost. ML potentials are now firmly established in the field of computational materials science and their application to homogeneous phases has been well documented.

In the present work, we describe a unified computational model for the Si–O system that we have obtained with the help of an active-learning scheme for local environments. We extract representative atomic environments from large-scale simulations and embed them in a melt-quenched amorphous matrix, allowing us to sample representative environments for the fitting of accurate ML potentials. Our final model shows high accuracy across a wide configurational space including high-pressure silica, silica surfaces, and mixtures of silica and silicon. We showcase the usefulness of the method by creating fully atomically resolved, 10-nm-scale structure models of amorphous and partially crystalline SiO.

## Results

### Active learning for nanoscale structure

We have developed a comprehensive dataset of atomistic structures and quantum-mechanical reference data for the binary Si–O system, as well as an interatomic potential fitted to that database in the atomic cluster expansion (ACE) framework[33–35]. We initialised the protocol with two existing datasets for silicon (Bartók et al.[30]) and silica (Erhard et al.[31]) respectively, and we then gradually explored the relevant configurational space using the active-learning workflow illustrated in Fig. 1. Quantum-mechanical reference ("training") data for energies and forces were obtained with the strongly constrained and appropriately normed (SCAN)[36] exchange–correlation functional for DFT, which shows good performance for elemental silicon[37] and the various silica polymorphs[31].

Our active-learning workflow follows three main tracks: high-pressure bulk silica, silica surfaces, and non-stoichiometric $SiO_x$ systems (Fig. 1a). The individual tracks are kept separate during initial training, i.e., they do not share their newly generated training data; however, in the end, all structures are merged into one comprehensive database.

The single subtracks are further divided into stages. In the first stage, we added initial structures, e.g., for crystalline high-pressure polymorphs or surface models. In the next stage, we fitted moment tensor potential (MTP) models[38] to the database and used these MTPs to explore configurational space in MD and to identify new structures by active learning[39]. Energies and forces for new structures were computed with DFT and added to the database. This process was iterated until the extrapolation threshold (Supplementary Note 1B) was not exceeded during the MD trajectories anymore.

The third stage, highlighted in red in Fig. 1a, is the most important part of our workflow, and is based on large-scale simulations in each track. We used 2–4 MTPs trained on the same database to estimate a per-atom committee error, as is commonly done for neural-network potentials[40]. For atoms with high uncertainty (Supplementary Note 1C), we extracted the environments into smaller, "DFT-sized" cells by an approach that we call amorphous matrix embedding (Fig. 1b). After identifying an atom with high uncertainty, we cut out a cube containing the corresponding environment of the atom. This cube has a size which is feasible for performing DFT computations; it is generally chosen larger than twice the cut-off of the potential. After extracting the cube, the atoms within the cut-off radius of the atom with high uncertainty are kept fixed. The remaining structure is melted in an ML-MD simulation to create an amorphous matrix and smooth boundaries. Details of the procedure can be found in Supplementary Note 1C. We note very recent, related approaches to isolating fragments for

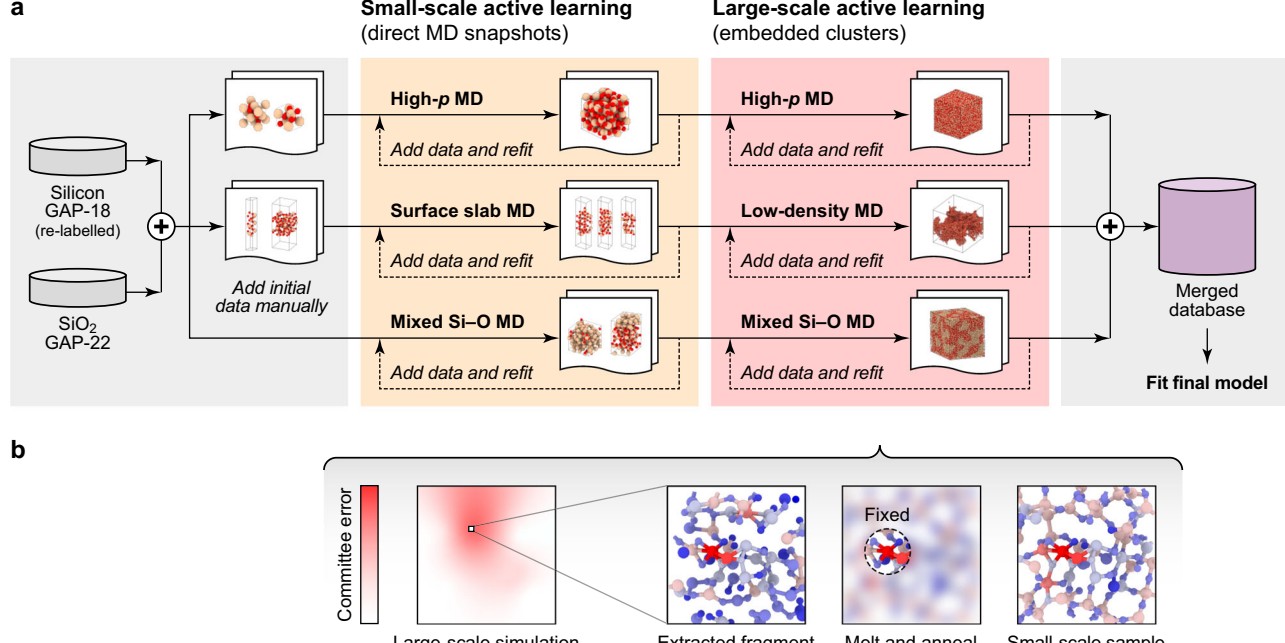

**Fig. 1 | An active-learning workflow for complex atomistic structures.**
**a** Overview of the procedure to obtain the database. After merging structures from the Si-GAP-18 (ref. 30) and SiO$_2$-GAP-22 (ref. 31) databases, the process was split into three tracks, aiming to describe high-pressure ("High-$p$") silica, silica surfaces, and mixed Si–O systems with different stoichiometric compositions. In each of the tracks, small-scale molecular dynamics (MD) simulations were used to sample new structures by active learning[38]. In the last step, large-scale simulations were performed, where atoms with high uncertainty were recognised by a committee error. **b** Schematic illustrating the concept of our amorphous matrix embedding

approach. First, we extract the wider environment of an atom with high uncertainty (indicated using a colourbar with red shading: note the "Large-scale simulation" is merely a schematic sketch). Then we keep the atom of interest, as well as the direct environment fixed, and we melt and anneal the outer environment (indicated by blurring the region outside the "Fixed" sphere). As a result, we obtain a small-scale structural model which has the atom of interest and its local environment embedded into an amorphous matrix. This sample can be fed into the training database.

active learning based on minimising the uncertainty for boundary atoms[41,42].

The final database was obtained by merging the data of all tracks together, including some additional samples such as clusters and vacancies. This database contains 11,428 structures with a total of ≈1.3 million atoms (Supplementary Table 1). For validation, we held out 5% of these structures from training, selected at random.

## Performance

The final potential is a complex non-linear ACE model, obtained by summation of one linear and seven non-linear ACE terms (Methods). This approach allows a more flexible description than just a linear or Finnis–Sinclair-like embedding, at only moderately higher computational expense. The resulting potential has a test-set root mean square error (RMSE) of 16.7 meV atom$^{-1}$ for energies and 306 meV Å$^{-1}$ for forces. These errors are averaged over the full dataset, however, and so they are not in themselves sufficient to characterise the quality of the potential. For example, they refer to a highly heterogeneous set of structures, with target energy values spanning more than 8 eV atom$^{-1}$, and a range of forces of 40 eV Å$^{-1}$ covered by the database. Furthermore, the numerical accuracy of the potential in certain parts of configurational space (e.g., crystalline polymorphs) is far more important than in others (e.g., liquid and amorphous structures).

In Table 1, we therefore show the performance of our model on different separate test sets. The complex non-linear ACE is compared to our previous silica GAP model described in ref. 31 ("SiO$_2$-GAP-22" in the following), and also to simpler ACE models fitted to the new database using linear and Finnis–Sinclair-like embeddings, respectively. Indeed, the complex non-linear ACE potential is the only one among the three which achieves comparable errors to SiO$_2$-GAP-22 for amorphous and crystalline silica structures. In contrast, for amorphous elemental silicon, mixed-stoichiometry as well as high-pressure phases the complex non-linear ACE is significantly more accurate than SiO$_2$-GAP-22, since these structures are not part of the GAP database. This table therefore indicates the main challenge – and its solution – in the present model compared to the previous GAP: both are highly accurate for crystalline (≈1 meV atom$^{-1}$) and bulk amorphous (≈5 meV atom$^{-1}$) SiO$_2$, but our ACE model caters to a much wider range of scenarios outside of the 1:2 stoichiometric composition.

Figure 2 shows the phase diagram of SiO$_2$ calculated by thermodynamic integration[43,44] using the ACE potential compared to a CALPHAD phase diagram from the literature[45]. The ACE and CALPHAD predictions agree well throughout, and for the boundary between quartz and coesite we observe almost quantitative

agreement. In contrast, the cristobalite and tridymite phases seem to be over-stabilised. At 0 GPa, the melting point is notably over-estimated (about 2400 K, compared to ≈2000 K experimentally[45]); moreover, the phase stability regions of both phases are more extended than in the reference. To illustrate the sensitivity of the analysis to small errors in predicted energies, we added a fictitious energy penalty of 5 meV atom$^{-1}$ for cristobalite and tridymite (Supplementary Fig. 1a); in this case, the transition lines agree much better with the CALPHAD reference than before. Further numerical tests showed that the tridymite–cristobalite transition line, in particular, is strongly affected by small shifts in energy (Supplementary Fig. 1b–f). We thus conclude that the quantitative deviation seen in Fig. 2 is due to the inaccuracy of the underlying exchange–correlation functional, rather than indicating a shortcoming of the ACE approach. This is an example of the more general problem that any issues with the ground-truth method (e.g., numerical instabilities) will translate into the ML model.

## High-pressure structural transitions of SiO$_2$

Figure 3 characterises high-pressure properties of silica. In Fig. 3a, we show energy–volume curves of α-quartz, coesite, stishovite, α-PbO$_2$-type, and pyrite-type silica as predicted by our ACE model and compared with DFT data, with which they agree well. In addition, we tested the behaviour of the model for rosiaite-type silica, which was recently observed in experiment[46] and predicted theoretically[47] for direct compression of α-quartz. In contrast to the structures mentioned before, this particular polymorph is not part of the training database. Nevertheless, the ACE model reproduces DFT data for this structure similarly well as for the other polymorphs.

Figure 3b shows an enthalpy–pressure diagram at 0 K. For lower pressures, there is a transition from α-quartz to coesite between 2.5 and 3.0 GPa, consistent with the predicted phase diagram (Fig. 2), followed by a transition to stishovite at 5.5–6.0 GPa. At higher pressures of ≈110 GPa, we observe the transition from stishovite to α-PbO$_2$-type silica. Experimentally, rather than stishovite (rutile type), the structurally closely related CaCl$_2$ (distorted rutile) type polymorph of silica is stable. The transition from CaCl$_2$- to α-PbO$_2$-type silica was observed at 120 GPa and 2400 K[48]. Given that our enthalpy data correspond to a temperature of 0 K, both values agree well with each other. For the transition of α-PbO$_2$- to pyrite-type silica, our ACE model predicts a pressure of ≈246 GPa, in good agreement with the experimentally determined transition pressure of ≈260 GPa at 1800 K[49]. Finally, rosiaite-type silica[46] is correctly identified as metastable over the pressure range studied.

## Table 1 | ML model performance

| | SiO$_2$-GAP-22 (ref. 31) | | Si–O ACE models (This work) | | | | | |
| | | | Linear (N = 1) | | Finnis-Sinclair-like (N = 2) | | Complex (N = 8) | |
| | ΔE | ΔF | ΔE | ΔF | ΔE | ΔF | ΔE | ΔF |
|---|---|---|---|---|---|---|---|---|
| SiO$_2$ crystals | 1.0 | 0.08 | 0.8 | 0.07 | 1.1 | 0.06 | 0.9 | 0.05 |
| a-SiO$_2$ (CHIK-MD) | 3.7 | 0.19 | 4.1 | 0.27 | 5.1 | 0.27 | 2.2 | 0.19 |
| a-SiO$_2$ (GAP-MD) | 1.1 | 0.10 | 10.3 | 0.13 | 9.8 | 0.12 | 4.6 | 0.10 |
| a-SiO$_2$ (ACE-MD) | 4.0 | 0.17 | 8.0 | 0.28 | 7.4 | 0.26 | 3.2 | 0.18 |
| a-SiO$_2$ surfaces | 14.9 | 0.18 | 21.4 | 0.21 | 18.0 | 0.18 | 4.7 | 0.16 |
| a-Si[a] | >1600 | >3.2 | 115.8 | 0.38 | 53.9 | 0.34 | 51.5 | 0.26 |
| a-SiO$_x$[a] | >4200 | >3.5 | 37.8 | 0.71 | 35.0 | 0.64 | 38.0 | 0.43 |
| high-p a-SiO$_2$[a] | 122.7 | 0.87 | 15.1 | 0.48 | 5.6 | 0.36 | 4.6 | 0.24 |

We report energy (ΔE) and force (ΔF) root mean square error (RMSE) values in meV atom$^{-1}$ and eV Å$^{-1}$ on different test sets. We characterise three ACE models, fitted to the same dataset but with increasing model complexity (Methods). 'a' indicates amorphous structures. The CHIK[25] and GAP[31] generated structures are taken from ref. 31. An extended version of this table with additional information can be found in Supplementary Table 3.
[a]Structural models generated using ACE-MD.

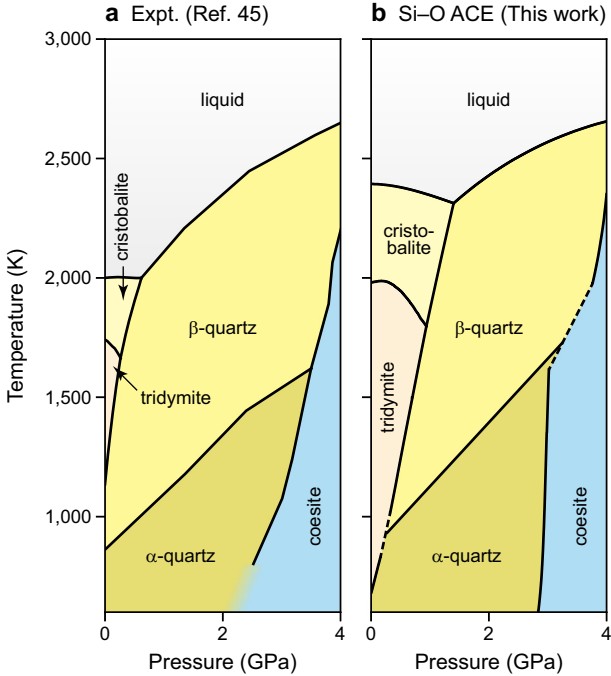

**Fig. 2 | Temperature–pressure phase diagram of SiO₂. a** Phase diagram calculated based on experimental data, adapted from the literature (ref. 45). **b** The same phase diagram calculated based on predictions from our Si–O atomic cluster expansion (ACE) model and thermodynamic integration. The background shading in different colours emphasises the stability regions of different phases. Source data are provided as a Source Data file.

Figure 3c shows the pressure evolution of the average coordination number (CN) of silicon atoms in amorphous silica, extracted from an MD simulation at room temperature and under isostatic pressure. The ACE results agree well with experiment up to about 50 GPa[5,50], and with ab initio MD[51] results over the whole pressure range. The good agreement with experiment is particularly pronounced for the data from ref. 50. Above 50 GPa, our model underestimates the average CN: at 175 GPa the experimental estimate is about 7; the ACE simulation predicts it to be 6. Importantly, this does not mean that there are no 7-fold coordinated environments, but there remain some 5-fold coordinated atoms as well, lowering the average (Fig. 3d). A possible reason for the good agreement with the ab initio result, but the deviation from experiment, might be the limited time scales in our simulations, which hinder a complete transition into higher-coordinated environments. Moreover, we note that computed X-ray-Raman spectra of the ab initio structures from ref. 51 are in good agreement with experiment indicating a lower CN. Other MD simulations also showed slightly lower CNs than the experimental values[52]. Figure 3e shows three different 7-fold coordinated environments extracted from the simulations.

A CN of 7 in amorphous silica might be surprising, since silicon is sixfold-coordinated in all crystalline silica polymorphs that are stable in this pressure range. However, the pyrite-type phase, which becomes thermodynamically stable at ≈240–260 GPa, contains silicon atoms with a 6+2-fold environment. A recent study found certain, but limited, similarities between these 7-fold environments in glassy and pyrite-type silica[52].

In Supplementary Fig. 2, we show two additional structural fingerprints which have been commonly analysed in experiment: the position of the first sharp diffraction peak and the Si–O bond length. For both cases, our simulations show good agreement with experiment.

## SiO₂ surfaces and aerogels

Figure 4 tests the ability of the potential to accurately predict surface energies. We begin with validation for different $\alpha$-quartz surfaces: we created surface slab models, relaxed them with the ACE potential, and evaluated the energetics, and therefore the surface energy per area, with DFT single-point computations (Fig. 4a). The ACE results agree well with DFT, especially considering that the training database does not contain all the surface terminations shown. The ACE model is also able to predict the stability of the reconstructed $\alpha$-quartz (001) surface (Supplementary Fig. 3a, b). Whilst these surface energies can be computed with DFT, realistic amorphous surface energies are much more difficult to obtain, due to the required system sizes. Therefore, Fig. 4b validates the potential on 125 small-scale surface structures of amorphous SiO₂, each containing 192 atoms. The surface models are created based on bulk structures from ref. 31; the latter had been generated in melt–quench simulations with different interatomic potentials and therefore span a range of energies. Regardless of the starting structure, the ACE model captures the surface energy for all slab models very well: the total RMSE is about 0.01 eV/Å², and only a slight underestimation compared to DFT is seen. Moreover, there are no clear outliers although the various surface energies indicate a large diversity of the surface structures.

The amorphous surfaces shown are already very complex, but in reality they are often not flat as here. They have curvature, for example when occurring inside pores, and such complex structures can no longer be directly validated with DFT. Figure 4c therefore shows how well atomic environments in various porous amorphous structures are covered by the dataset. These structures were prepared by straining amorphous structures at elevated temperatures to the desired density. To validate the performance of the potential on this model, we show the linear extrapolation grade according to the maxvol selection[41,53]. An extrapolation grade above 1 corresponds to atomic environments that have not been covered in the training database. This does not mean that the potential is no longer reliable, as there is a certain range of more or less reliable extrapolation, but as the extrapolation grade increases, non-physical behaviour and failure of the potential becomes more and more likely[39]. For all porous structures, regardless of density, we find that the maximum extrapolation value is less than 1. Thus, we observe no pronounced extrapolation in any of the cases considered, indicating an accurate description of the potential for a variety of curved surfaces. In Supplementary Fig. 3e, f, we demonstrate the effectiveness of the ACE potential in reconstructing artifical amorphous surfaces. To this end, we cut out a spherical cavity from an amorphous bulk structure. Upon heating, the amount of incorrectly arranged atoms near the surface (<3 Å) decreases over time, indicating that the surface atoms are undergoing significant reconstructions.

## Elemental silicon

Whilst our ACE model is designed for the binary Si–O system, we show in Table 2 the performance for diamond-like elemental silicon compared to both DFT and experiment. The bulk modulus is very well reproduced, whereas the vacancy formation energy is underestimated by about 30%. The experimental surface energies are well recovered by the ACE model, but this may be partly due to serendipity, because the SCAN ground-truth data show poorer quality (Table 2). Moreover, the potential captures the reconstruction of the diamond-type silicon (100) surface (Supplementary Fig. 3c, d). We also computed the linear thermal expansion coefficient of diamond-type silicon in the quasi-harmonic approximation, finding almost perfect agreement between the ACE prediction and experiment; in particular, the unusual negative expansion coefficient below 130 K is reproduced (Supplementary Fig. 4). Finally, melt–quench simulations were performed to generate a-Si structures (Supplementary Fig. 5). The agreement with the experimental structure factor is as good as that for a GAP-18-generated

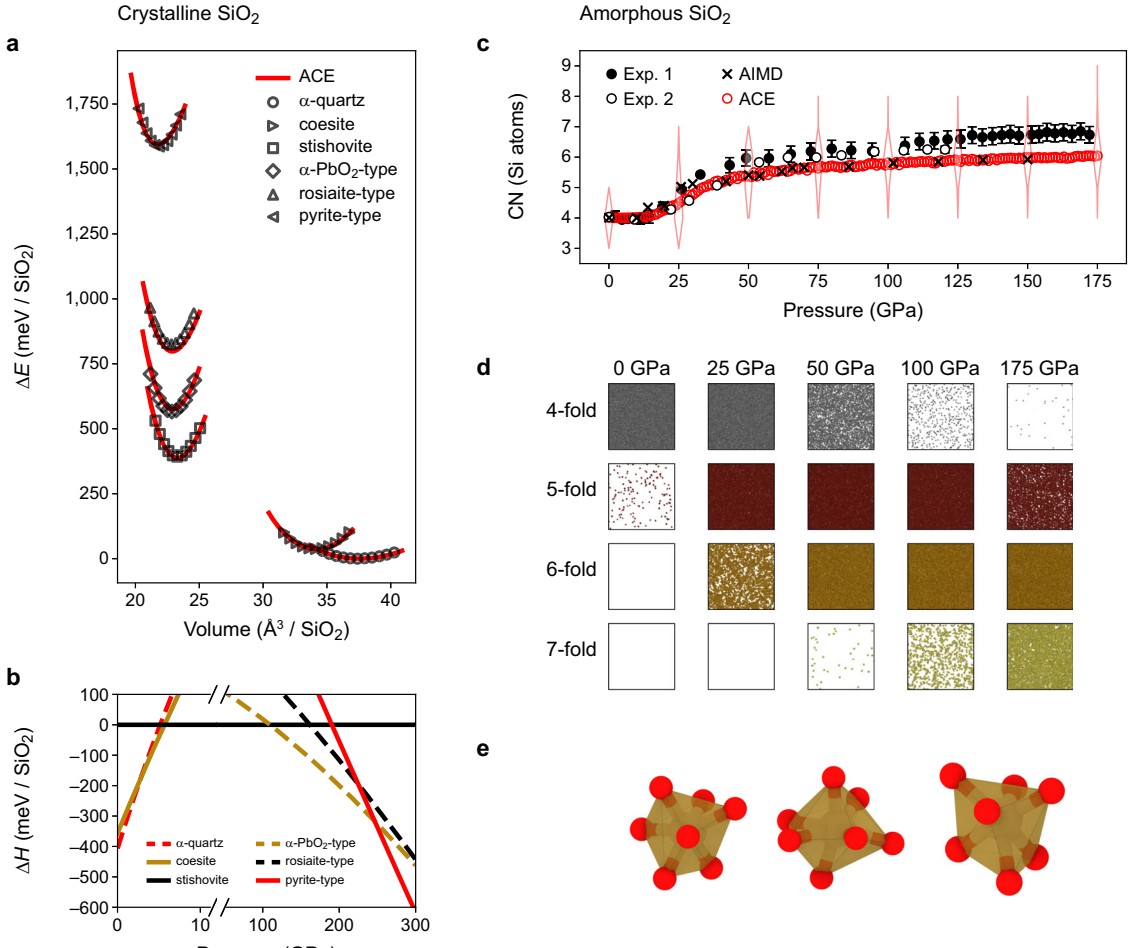

**Fig. 3 | Silica at megabar pressures. a** Energy–volume curves for high-*p* silica polymorphs with energies referenced to *α*-quartz (Δ*E*). Markers indicate results from density-functional theory computations with the strongly constrained and appropriately normed exchange–correlation functional (SCAN DFT); lines indicate the atomic cluster expansion (ACE) predictions. **b** Enthalpy differences (Δ*H*) for various silica polymorphs referenced to stishovite. **c** Compression of a vitreous silica structure. Results for the average silicon coordination number (CN) are compared to experimental measurements from ref. 5 ("Exp. 1") and ref. 50 ("Exp. 2") and ab initio MD (AIMD) simulations from ref. 51. The distribution of coordination numbers at selected pressures is indicated by violin plots. **d** Snapshots of the compression simulation showing coordination polyhedra for different coordination numbers (only). **e** Visualisation of the coordination environments of selected 7-fold coordinated silicon atoms. Source data are provided as a Source Data file.

structure from ref. 54. In addition, we are able to achieve lower quenching rates with the ACE than with the GAP, and for quenching rates as low as $10^{10}$ K s$^{-1}$, we observed crystallisation.

Compared to Si-GAP-18[30], we observe higher errors with respect to the reference data (Table 2). This is not a principal shortcoming of ACE compared to GAP: it was already shown that it is possible to fit an ACE potential with similar numerical accuracy as Si-GAP-18 to the same training database[34]. Instead, the lower accuracy might be caused by the extension of the database to a second element, compared to the Si-GAP-18 one, and its strong focus on the $SiO_2$ part of the configurational space. We assume that this causes, in turn, a less accurate description of the configurational space of the elemental species. Larger training databases might help to overcome this issue in the future. Due to the lower accuracy in reproducing the SCAN data, our potential has some shortcomings for higher-pressure structures: the bc8 phase is erroneously predicted to be stable at elevated pressure (Supplementary Fig. 6), and upon compressing a-Si we do not observe the eventual crystallisation that is described by Si-GAP-18 (Supplementary Fig. 7)[55]. We emphasise that very-high-pressure silicon phases were not the scope of the present work – instead, we focus on the accurate description of ambient-pressure silicon as a constituent part of mixed binary phases and nanostructures.

## SiO and mixed silicon–silica systems

Whilst the results so far have served to demonstrate the usefulness of the approach – both in terms of development of datasets and the fitting within the ACE framework – we are now able to study an actual application problem. To this end, Fig. 5a shows structural models of SiO. Experimentally, amorphous SiO is obtained by deposition of SiO from the gas phase[56]. In contrast, we created our models by melt–quench simulations. SiO phases are known to be metastable with respect to Si and $SiO_2$. For example, a recent DFT-based crystal-structure prediction study explored possible ordered phases of homogeneous SiO, and found that these are metastable compared to a mixture of crystalline Si and $SiO_2$[57]. We verified that our ACE potential similarly reproduces the metastability of the ambient-pressure phases (Supplementary Fig. 9), and that it accurately predicts relevant Si–$SiO_2$ interface energies (Supplementary Fig. 10). In good agreement with these results, our melt-quenched structures show a clear segregation between a-Si-like (blue) and a-$SiO_2$-like regions (red). With decreasing quench rate, the number of silicon grains decreases while their size increases. Figure 5b shows the structure factor, $S(q)$, for the structure quenched with $5 \times 10^{12}$ K s$^{-1}$ (data for the other structures are shown in Supplementary Fig. 11), which agrees best with the experimental data from ref. 15. Figure 5c shows the ratio between the volume of the

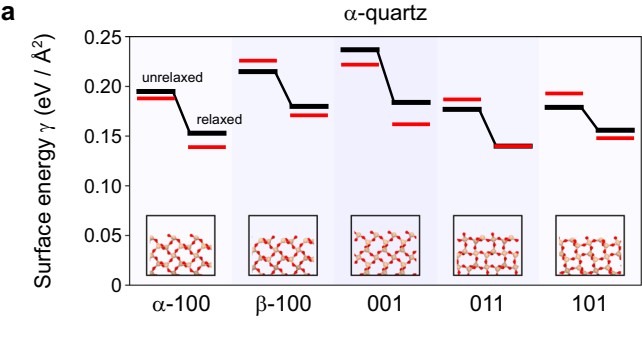

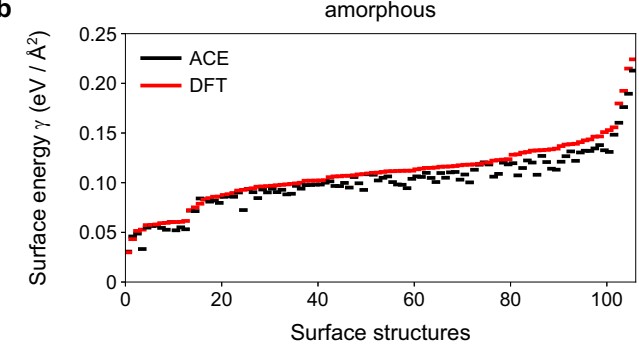

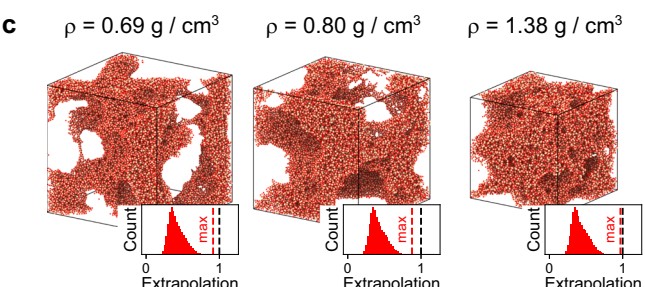

**Fig. 4 | Surfaces and aerogels. a** Surface energies $\gamma$ of various $\alpha$-quartz surfaces in relaxed and unrelaxed state. The black lines show the atomic cluster expansion (ACE) results while the red lines show the density functional theory (DFT) results for the unrelaxed structures and DFT re-relaxed structures. The structure pictures show the unrelaxed surfaces. **b** Surface energies of amorphous models calculated with ACE and DFT. Amorphous structures were taken from ref. 31, and have been created using various interatomic potentials: the BKS[23], CHIK[25], SiO2-GAP-22[31], Munetoh[59], and Vashishta[24] models. The structures were relaxed with ACE and sliced at various points, followed by another ACE relaxation. DFT surface energies were determined without further relaxation. **c** Exemplary porous amorphous silica structures with various densities $\rho$. Additionally, we show the distribution of the linear extrapolation grade[41] and the maximum extrapolation grade (red line) for each structure. For all structures, the maximum extrapolation grade is below one. Source data are provided as a Source Data file.

silicon grains divided by the interface area. In the approximation of spherical particles, the grain diameter is $d = 6 \cdot V_{\text{Si,grains}}/A_{\text{interface}}$. From this we can estimate average grain diameters between 24 and 54 Å for the tested quench rates. These grain diameters agree very well with transmission electron microscopy measurements, which indicated diameters of 30–40 Å[14].

Figure 5d shows the excess energies of the structures referenced to to $\alpha$-quartz and diamond-type silicon. The SiO structures were relaxed by optimisation of the cell size as well as the atomic positions at 0 K. As experimental reference, we show the standard enthalpy of formation of SiO[58]. The structures generated using quench rates of $5 \times 10^{12}$ and $2 \times 10^{12}$ K s$^{-1}$ have energies comparable to experiment. Indeed, we can even create structures that are energetically more favourable than in experiment, noting again that our procedure to produce the structures deviates significantly from the experimental one.

But is this really an improvement compared to existing, empirically fitted interatomic potentials? We tested the Munetoh potential[59] and a charge-optimised many-body (COMB) potential[29] for the same procedure to generate structural models of SiO. The Munetoh potential yielded a homogeneous structure without observable segregation into silicon and SiO2, and the resulting structure factor (Supplementary Fig. 11c) deviates strongly from experiment. For the COMB potential, we observed pore formation at elevated temperatures, finally resulting in a strongly increased simulation-cell size. Therefore, we only equilibrated our best-matching structure at room temperature and analysed the change in structure factor (Supplementary Fig. 11f): again, we observed a strong deviation from experiment, indicating that the structure is very different from the ACE model prediction.

## Crystallisation of silicon in amorphous SiO

Silicon in silicon monoxide is experimentally known to crystallise above 850 °C[60]. Figure 6 illustrates simulations of such crystallisation processes. The SiO structures shown in Fig. 5 were heated to 1400 K, causing the silicon-rich regions to melt while the silica matrix remained solid. The structures were then quenched to 1200 K within 20 ns (Fig. 6a). Through this cooling process, we noticed crystallisation in the silicon-rich regions of the SiO structure. Details are shown in Supplementary Fig. 12, indicating how crystallisation starts from two seeds, appearing shortly after each other and propagating throughout the structure. Before thermal treatment, all structures show nearly no sign of crystallinity, whereas afterwards, the structures with larger silicon grains do (Fig. 6c–f). This also affects the structure factor: those systems with no or only a small amount of crystallinity (quench rates: $1 \times 10^{13}$ and $5 \times 10^{12}$ K s$^{-1}$) show structure factors that are still similar to the experimental structure factor of SiO (Fig. 6g), whereas those with larger amounts of crystalline silicon ($1 \times 10^{12}$ and $2 \times 10^{12}$ K s$^{-1}$) show distinct $S(q)$ peaks that indicate crystallinity (Fig. 6h). This comparison allows us to exclude the occurrence of large amounts of crystalline silicon in silicon monoxide samples, given the experimental $S(q)$ (cf. Fig. 6g and ref. 15).

Figure 6i shows that all structures are energetically more favourable after the thermal treatment. However, even though one might expect that the stronger crystallised structures gain more energy, we observe no direct connection between the energy gain and the level of crystallinity after heat treatment. The reason is likely that for the fast-quenched structure ($1 \times 10^{13}$ K s$^{-1}$), the silicon-rich regions might be not perfectly arranged. Thermal treatment thus lowers the interface energy and the internal energy of the amorphous silicon region. For the crystallised structures there is, on one hand, an energy gain due to crystallisation but, on the other hand, an energy loss due to the higher interface energy between crystalline silicon and the silica matrix compared to that of a-Si and the silica matrix (cf. Supplementary Fig. 13). Based on these interface energies, we constructed a simple physically-based model (Methods) to calculate the energy gain of a spherical silicon inclusion in an amorphous silica matrix by crystallisation. This energy gain is shown in Fig. 6j for interface energies taken from four pairs of manually constructed interfaces. The difference between the models is rather small, indicating that the minimum radius of a spherical inclusion has to be around 9 Å to be energetically favourable in a crystalline state. We note that the manually constructed models do not have the same interface orientations as in the SiO structures and are also not perfectly relaxed, causing an overestimation of the interface energies – as seen, for example, in the difference between the interface energies of the SiO structures and of the manually constructed a-Si–a-SiO2 interface in Supplementary Fig. 13d. However, since for our model the difference between the interface energy between a-Si–a-SiO2 and

c-Si−a-SiO$_2$ is the only relevant quantity, we assume that these effects partially cancel.

## Discussion

Understanding the microscopic nature of interfaces and nanostructured matter is essential to advancing materials research. Here, we have presented an active-learning scheme that we term "amorphous matrix embedding" that can realistically represent environments from large-scale simulations in DFT-accessible cells, enabling fast and accurate atomistic modelling of heterogeneous materials. We used the approach to develop a general-purpose interatomic potential for binary Si−O phases with varied compositions that is able to describe the trifecta of modelling challenges in this material system: very-high-pressure phases (relevant to geology), surfaces (relevant to catalysis), and mixed stoichiometric compositions with nanoscale heterogeneity (relevant to battery systems).

Using the ACE approach, we observe a speed-up of about two orders of magnitude compared to the more established GAP framework. This makes it possible to access long time scales and large length scales with DFT-like accuracy. Of course, there are still some shortcomings of this potential, e.g., the lower accuracy for pure silicon compared to the state of the art – but this use-case is not the focus of our work, as there are already competitive GAP and ACE models available[30,34]. In our case, the quality of the underlying meta-GGA data might cause an outperformance compared to earlier ML-potentials fitted with more economical GGA labels. The potential also underestimates the FSDP height of amorphous SiO$_2$ (Supplementary Fig. 14), as already observed for our earlier GAP model[31]. Further research is required to identify the origin of this underestimation.

We hope that our work, and the dataset and resources developed therein, will advance the modelling of porous silica nanostructures as well as of high-pressure silica. For the Si−SiO$_2$ interface, alternative interatomic potential models are scarce and the higher-quality potentials come with an expensive charge-equilibration term. Our tests showed that the ACE potential describes silicon monoxide in much closer agreement with experiment than existing empirical models. Additionally, we are able to generate crystallites in the SiO matrix as observed experimentally, showing that the ACE potential can be used for a wide range of applications in the Si−O system, including both ordered and disordered structures.

We view the present database and ML potential model as a starting point for wider-ranging studies in this important material system. In the future, higher accuracy for the mixed system might be achieved by using charge-equilibration schemes coupled with ML potentials[61]. However, this would come with much longer computing times as well as worse scaling for larger systems. Moreover, in the future, we will include lithium in the potential to investigate the battery performance of SiO on the atomistic scale.

## Methods

### Machine-learning potential fitting

We used two frameworks for fitting ML potential models. While constructing the reference database, we used Moment Tensor Potentials[38] with active learning[53] as implemented in the `MLIP` package[39]. For the

### Table 2 | Properties of diamond-type silicon

| Property | | This work | | Ref. 30 | | |
|---|---|---|---|---|---|---|
| | | SCAN | ACE | PW91 | GAP | Expt. |
| Bulk modulus | (GPa) | 100.0 | 100.8 | 88.8 | 88.4 | 97.8[71] |
| Vacancy $E_f$ | (eV) | 4.09 | 2.80 | 3.67 | 3.61 | 4[72] |
| $\gamma_{100}$ | (eV/Å$^2$) | 0.155 | 0.116 | 0.135 | 0.133 | 0.133[73] |
| $\gamma_{110}$ | (eV/Å$^2$) | 0.126 | 0.089 | 0.095 | 0.094 | 0.094[73] |
| $\gamma_{111}$ | (eV/Å$^2$) | 0.113 | 0.075 | 0.098 | 0.096 | 0.077[73] |

We show values from density-functional theory (DFT) computations with the strongly constrained and appropriately normed (SCAN) exchange–correlation functional for reference, as well those obtained with the complex atomic cluster expansion (ACE) potential; both computations are compared to the Si-GAP-18 Gaussian approximation potential model and the corresponding DFT reference data (PW91), taken from ref. 30, and to experimental data ("Expt."). $E_f$ is the vacancy formation energy. $\gamma_{100}$, $\gamma_{110}$ and $\gamma_{111}$ are the surface energies of the diamond-type silicon (100), (110), and (111) surfaces, respectively.

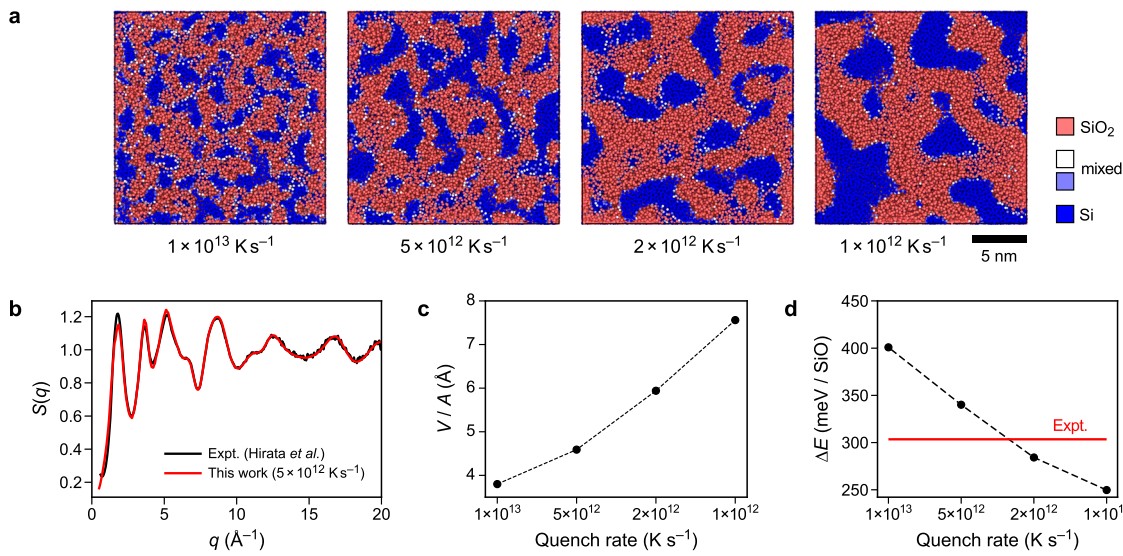

**Fig. 5 | Nanoscale segregation in amorphous silicon monoxide. a** Visualisation of SiO structures generated by quenching from the melt at rates between $10^{13}$ and $10^{12}$ K s$^{-1}$. Colour-coding is based on the nearest-neighbour count up to 2.0 Å, taken to correspond to the Si–O coordination numbers, which are four in SiO$_2$ and zero in elemental silicon. Accordingly, SiO$_2$-like and Si-like regions are indicated in red and blue, respectively. **b** Structure factor, $S(q)$, as a function of the wave vector, $q$, for the $5 \times 10^{12}$ K s$^{-1}$ simulation, determined at 300 K. **c** Relation between grain volume, $V$, of the silicon grains and the interface area, $A$, between silicon and silica. With increasing quench rate, the grain size of the structures decreases. **d** Energy of the SiO structures referenced to α-quartz and to diamond-type silicon ($\Delta E$), compared to the experimental standard enthalpy of formation for SiO[58]. The predictions shown are robust with respect to system size (Supplementary Fig. 8). Source data are provided as a Source Data file.

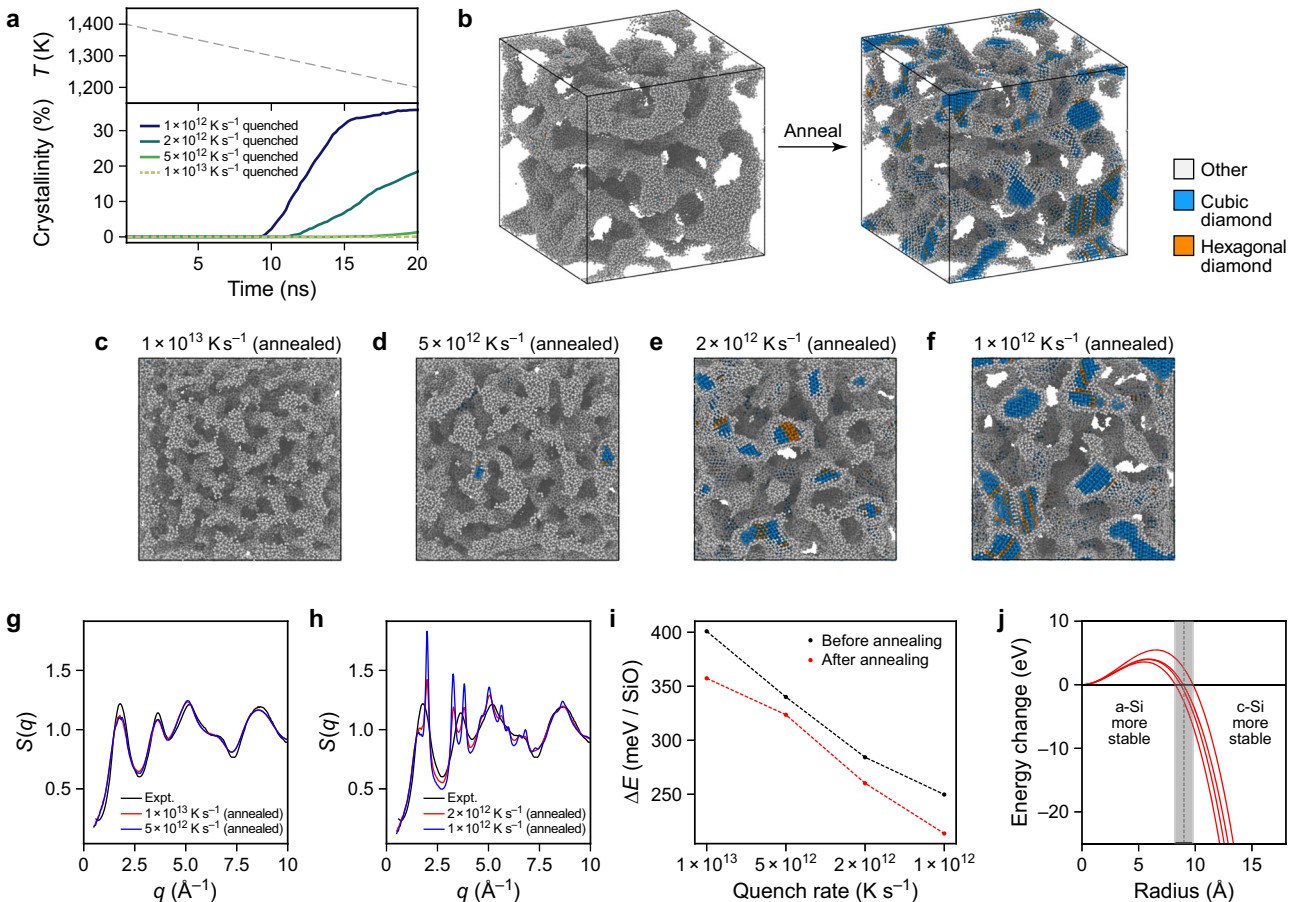

**Fig. 6 | Crystallisation of silicon in silicon monoxide. a** Evolution of crystallinity over time for the thermal treatment of the different SiO structures. The temperature ($T$) change is the same in all cases, and characterised in the upper panel. **b** The silicon network within the SiO structure quenched at $10^{12}$ K s$^{-1}$: for this image, we only show Si atoms which are not bonded to more than one O atom (2 Å cut-off). The colour coding indicates the type of structure recognised by polyhedral template matching (root mean square deviation cut-off: 0.1). The initial structure is completely amorphous; after thermal treatment, crystalline regions are observed. **c–f** All structures from Fig. 5a after thermal treatment. **g, h** X-ray structure factors

$S(q)$ as function of the wave vector $q$ for the structures shown in panels **c–f**. For the structures with high crystallinity, the structure factor contains more distinct peaks. **i** Computed energy of formation $\Delta E$ for structures before and after heat treatment. **j** Energy change of a spherical silicon inclusion within an amorphous silica matrix by crystallisation. We used interface energies from manually constructed interface structures generated using amorphous models from four different quench rates. Therefore, we show four separate data series as lines. Source data are provided as a Source Data file.

final potential fit, we used the nonlinear Atomic Cluster Expansion (ACE)[33,34] as implemented in PACEMAKER[35]. For ACE, we tested a range of combinations of embeddings, and found the following to be suitable:

$$E_i = \phi + \sqrt{\phi} + \sum_i \phi^{f_i}, \qquad (1)$$

with $\phi_i$ being atomic properties, which are expanded by the ACE basis functions (for details see ref. 33). The exponents of the embeddings include fractional exponents and higher integer powers of $f_i \in \{1/8, 1/4, 3/8, 3/4, 7/8, 2\}$. We found that especially fractions between 0 and 1 improved the behaviour of the potential. This approach goes beyond the previously suggested linear embedding (only the first term) and Finnis–Sinclair (the first two terms) type embedding[34], and is referred to as "complex" embedding in Table 1. For the expansion of the atomic properties $\phi_i$ we used 600 basis functions with 5700 parameters. As radial basis we employed Bessel functions. A $\kappa$ value of 0.01, which gives the ratio between force and energy weights value, was used during fitting. For optimisation we used the BFGS algorithm for 2000 steps.

## DFT computations

All DFT computations were performed using VASP[62,63] and the projector augmented-wave method[64,65]. We used the SCAN functional[36] with an energy cutoff of 900 eV and a $k$-spacing of 0.23 Å$^{-1}$. Surface were performed with dipole corrections. We note that these convergence parameters are optimised for silica; however, we found them to be also well converged for mixed phases and for pure silicon structures. Only for very-high-pressure silicon allotropes, a higher $k$-spacing would provide a relevant advantage; however, since these are not in the scope of the present work, we neglect these inaccuracies.

## MD workflows

Simulation protocols were implemented using the atomic simulation environment (ASE)[66] and the OVITO Python interface[67]. While optimisation and small-cell MD were partially performed with ASE, large-scale MD and statics simulations were carried out using LAMMPS[68]. The time step was 1 fs. For NVT simulations, we used a Nosé–Hoover thermostat with temperature damping constant of 100 fs; for NPT simulations, we added a Nosé–Hoover barostat with a pressure damping constant of 1000 fs.

For quench simulations we used the same protocol as in ref. 31. This protocol starts with a randomisation part at 6000 K for 10 ps (NVT). Afterwards the temperature is immediately reduced to 4000 K and held there for 100 ps (NPT, zero external pressure). From there the melt is quenched with different quench rates to 300 K (NPT, zero external pressure). At this temperature the structure is equilibrated for another 10 ps. In case of 'hybrid' simulations these quenches have been performed using the CHIK potential[25] and afterwards the structure has been equilibrated for another 20 ps with the ACE potential.

As input for the compression simulations we generated amorphous structures using this quenching protocol with a quench rate of $10^{13}$ K s$^{-1}$ and only the ACE potential. The compression was performed under isostatic conditions. In each step, the pressure was initially increased by 1 GPa within 2.5 ps of simulation time, followed by equlibration over 2.5 ps at the new pressure. This procedure was iteratively repeated. Coordination numbers were determined after equilibration.

The aerogel structures were created by a similar protocol as in ref. 31. An initial structure was randomised at 6000 K for 10 ps, instantly cooled to 4000 K and kept there for 100 ps. From this temperature, the liquid was cooled to 300 K with a quench rate of $10^{13}$ K s$^{-1}$. During the equilibration at 4000 K and up to half of the quenching process, the cells were additionally extended to the desired density.

The mixed structures were created using the same protocol as in ref. 31 for producing amorphous structures. The volume of the silicon grains was determined within OVITO by creating bonds between silicon and oxygen atoms (cut-off: 2 Å) and deleting all atoms which have more than one such bond. This deletes the whole silica matrix. The interface area and volume is then determined by using the `ConstructSurfaceMesh` modifier (Gaussian density method, resolution: 50, radius scaling: 100%, isovalue: 0.1) on the remaining atoms.

### Phase diagram calculations

Thermodynamic integration was carried out as implemented in `calphy`[43,44]. We used 50,000 equilibration steps, 800,000 switching steps for the switching to the Einstein crystal, as well as 300 steps/K for the thermodynamic integration to calculate the temperature dependence. Due to numerical issues, we fixed the spring constants of the Einstein crystal to 2 eV Å$^{-2}$ for oxygen and 4 eV Å$^{-2}$ for silicon. We carefully checked the influence of this constraint on the final results and found it to be negligible. More details on the phase diagram calculations can be found in Supplementary Note 2.

### Structure factors

Faber–Ziman structure factors were obtained by summation of the Fourier transformations of the partial radial distribution functions calculated with OVITO. The corresponding partial structure factors were weighted by atomic form factors taken from ref. 69. For the high-pressure structures, we used a cut-off radius of 20 Å for the radial distribution function, and analysed a single snapshot (without time averaging). For the SiO structures, we used a cut-off of 80 Å and an average over 10 snapshots.

### Surface energies

In these calculations we consider only stockiometric slabs. The surface energies, $\gamma$, were calculated as

$$\gamma = \frac{E_{slab} - N \cdot E_{ref}}{A}, \tag{2}$$

where $A$ is the total surface area (at the top and bottom of the slab combined), $N$ is the number of particles in the slab, $E_{ref}$ is the bulk reference energy and $E_{slab}$ is the potential energy of the slab. The slab energies for Table 2, Fig. 4a (relaxed), and Supplementary Fig. 3 have been calculated for DFT- and ACE-relaxed structures. For Fig. 4b, we used the ACE-relaxed structures also to determine the DFT single-point surface energy. The reference energy for the $\alpha$-quartz surface is the energy of the optimised $\alpha$-quartz unit cell per atom, and the reference energy for the diamond surface is obtained for the optimised diamond-type unit cell. The reference energy of the amorphous sample is the bulk energy of the same relaxed amorphous structure without surfaces.

### Enthalpy and structural analysis at high pressure

The enthalpy $H$ is given by

$$H(p) = E(V) + p(V) \cdot V, \tag{3}$$

where $E$ is the internal energy, $p$ is the pressure, and $V$ is the volume. The volume dependence of the energy was determined by a Birch–Murnaghan fit to the energy–volume curve of each polymorph. $p(V)$ was given by the corresponding derivative. The energy-volume curves were calculated by varying the volume by $\pm 20\%$ for $\alpha$-quartz and coesite, by $\pm 25\%$ for stishovite and $\pm 30\%$ for all other phases. The corresponding structures were structurally optimised, allowing changes of the positions as well of the box shape, however keeping the volume fixed. For the analysis of the compression MD simulations, coordination numbers were determined by integrating over the first peak of the partial Si–O radial distribution function. The Si–O bond distances are given by the first peak position of the partial Si–O radial-distribution function.

### Interface energy model for the SiO crystallisation

We build a interface energy based model for a spherical inclusion of silicon with radius $r$ within an amorphous SiO$_2$ matrix. The energy difference between the crystallised silicon and the amorphous silicon is given by,

$$\Delta E = 4\pi r^2 (\gamma_{c-Si-a-SiO_2} - \gamma_{a-Si-a-SiO_2}) - \frac{4\pi r^3 (E_{a-Si} - E_{c-Si})}{3V_{atom}}. \tag{4}$$

Here, $\gamma_{a-Si-a-SiO_2}$ is the interface energy between amorphous silicon and amorphous silica, $\gamma_{c-Si-a-SiO_2}$ is the interface energy between crystalline silicon and amorphous silica, $E_{a-Si}$ is the energy of the corresponding amorphous silicon structure (cf. Supplementary Table IV), $E_{c-Si}$ is the energy of crystalline silicon, and $V_{atom}$ is the volume per atom. Details about the interface energies we used can be found in Supplementary Note 2.

### Reporting summary

Further information on research design is available in the Nature Portfolio Reporting Summary linked to this article.

## Data availability

The potential parameter files, the reference data with SCAN labels, and additional supporting data (including `LAMMPS` scripts and input configurations) generated in thus study are openly available in the Zenodo repository at https://doi.org/10.5281/zenodo.10419194[70]. Source data are provided with this paper.

## Code availability

The codes for potential fitting and evaluation are publicly available and were used as provided by their respective authors, without modification. Custom-written Python scripts for data analysis are provided together with the corresponding data in the Zenodo repository.

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

## Acknowledgements

L.C.E. thanks Niklas Leimeroth for useful discussions. L.C.E. acknowledges support from the German Academic Exchange Service (Forschungsstipendien für Doktorandinnen und Doktoranden) and the Erasmus+ programme for support of two research stays at the University of Oxford. The research was supported by the Bundesministerium für Bildung und Forschung (BMBF) within the project FESTBATT under Grant No. 03XP0174A. J.R. and K.A. acknowledge support by the Deutsche Forschungsgemeinschaft (DFG, Grant no. 405621137, 405621160, and 521536863). L.C.E. acknowledges helpful discussion within the DFG GRK-2561 MatCom-ComMat. The authors gratefully acknowledge the computing time provided to them at the NHR Center NHR4CES at TU Darmstadt (project number 01539 and p0020142). This is funded by the Federal Ministry of Education and Research, and the state governments participating on the basis of the resolutions of the GWK for national high performance computing at universities (www.nhr-verein.de/unsere-partner).

## Author contributions

L.C.E. performed all computations and analysis, with guidance from J.R., K.A., and V.L.D. All authors contributed substantially to the design of the research and to the interpretation of the results. L.C.E. and V.L.D. wrote the paper with input from all authors.

## Funding

## Competing interests

The authors declare no competing interests.
