## [Peer Review File · Nature Communications]

Modelling atomic and nanoscale structure in the silicon-oxygen system through active machine learningREVIEWER COMMENTS

Reviewer #1 (Remarks to the Author):

The authors report the state of the art of machine learning technique applied to a series of Si–O system. Although underlying technique of the paper is already reported in ref 31, I positively recommend the ms. for publication in Nature Communications with minor revisions because the introduction of large-scale active learning is quite important and the application to a wide range of Si–O system including amorphous silicon monoxide (please see ref. 15; Hirata, A. et al. Atomic-scale disproportionation in amorphous silicon monoxide. Nat. Commun. 7, 11591 (2016)) is very impressive. I believe that the contents are suitable for a wide range of broad readers of Nature Communications because Si–O system is very important as the authors mentioned in the introduction part of the ms.

1. I recommend the authors to refer the following article because this article reported the landmark study on the structure of amorphous silicon monoxide, although the content is controversial.

Greaves, G. N. EXAFS and the structure of glass. J. Non-Cryst. Solids 71, 203–217 (1985).

2. I strongly recommend the authors to calculate neutron structure factor for amorphous silica because neutrons are sensitive to oxygen atoms. This is helpful supplemental data for a benchmark. Numerical data is available at the following web site.

<http://www.alexhannon.co.uk/DBindex.htm>

3. The authors may be interested in the following article.

Onodera, Y. et al., Understanding diffraction patterns of glassy, liquid and amorphous materials via persistent homology analyses. J. Ceram. Soc. Jpn. 127, 853–863 (2019).

Reviewer #2 (Remarks to the Author):

The authors present the development and application of a machine-learned interatomic potential for Si-O systems. The results are scientifically sound and the paper is written in a compelling and thorough way that makes it easy to follow and enjoyable to read. While I am not 100% sure that the novelty of the work compared to previous works justifies the publication in such a high-impact journal, the importance of the material system combined with the timeliness of the approach does. For this reason, I recommend publishing this paper.

Comments:

- To claim that the potential is suitable to study surfaces, I think it is critical to show that surface reconstructions and relaxations are properly captured. Fig. 4a only shows DFT single-point results for the unrelaxed and relaxed ACE structures. How far off are the relaxed DFT values? Moreover, does the potential capture reconstructions such as six-membered rings for the α -quartz (001) surface [<https://doi.org/10.1039/B701176H>] or the 2x1 reconstruction for (001) Si surfaces?
- Some simple tests in the spirit of Fig. 4a for SiO₂/Si interface energies would strongly support that the segregation of SiO₂ and Si is properly captured by the potential.
- A stunning result obtained with the GAP-18 for silicon is that it is able to describe the polymorphic transition from LDA-Si to HDA-Si – even though no high-pressure amorphous structures were included in the fit data base [<https://doi.org/10.1038/s41586-020-03072-z>]. Moreover, the GAP-18 potential correctly describes E(V)-curves for the high-pressure st12 and bc8 phases [<https://doi.org/10.1103/PhysRevX.8.041048>], in contrast to the ACE potential. Since the GAP-18 data base is part of the data base for the present potential, but the major shortcoming of the present potential is that high-pressure phases of pure Silicon are poorly described, I wonder where this loss of transferability comes from. Is this a consequence of the GAP vs ACE framework

or does the other fit data impede the transferability? Does the current potential capture the polymorphic transition? In my opinion, a more thorough discussion of this aspect might help other researchers in the quest of designing truly general-purpose potentials.

- How do the results of Fig. 5 depend on the system size?

- Ref. 3 in the SM " $\{\mathrm{SiO}\}_2$ " -> SiO₂

- The "construct surface mesh" modifier results in OVITO should depend on the probe sphere radius. For the sake of reproducibility, I recommend adding this parameter to the methods section.

Reviewer #3 (Remarks to the Author):

The authors present an ACE Si-O model trained using active learning on DFT(SCAN) reference data. One of the prominent contributions is that the reported ML potential is two orders of magnitude faster than GAP model at the expense of a small reduction in accuracy. The article presents a thorough analysis of the materials' properties using such model. In general, it provides good results, making it of interest for the materials' science users. Nevertheless, this work only focuses on the analysis of the models' characterization and not on a specific problem that, for example, reveals a new understanding on the dynamics of Si-O material.

Some further comments are presented below:

-When citing the supplementary material, please state the specific section.

-Table I is a good overview of the accuracy, but what is more informative to assess accuracy are the forces, this because MD simulations only used forces to sample the phase space. Hence, I would recommend to also include force errors in this table.

-Fig. 2 Phase diagram: Here, we can see that qualitatively, the ACE model can recover the phase diagram, nevertheless, it shows a severe deficiency predicting the transition liquid <-> solid of ~400K in the 0-1GPa region. This is a common issue in ML potentials trained on undersampled liquid phases. Meaning, the ML mostly learn the solid phases. The same happens with water models, where the simulations tend to generate icy phases instead of the proper liquid state.

-The article presents an interesting manual-fine tuning of the model via manipulating the different penalties on the loss function for the different phase states. This corrects some of the boundaries in the phase diagram. This could be a valid non-machine-learned way to compensate undersampled phases. Although, a full analysis on the rest of physical properties is required to assess if the model continues to be physically valid.

-The authors stated: "We thus conclude that the quantitative deviation seen in Fig. 2 is due to the inaccuracy of the underlying exchange–correlation functional, rather than indicating a shortcoming of the ACE approach." In general, a more elaborated note should be added to discuss the accuracy of the reference DFT data, this because it is well known that, even though the SCAN functional provides accurate energies, its gradients (i.e. forces) are problematic and highly depend on the stability of the implementation. Thereby, this could be a potential source of predictive error beyond the ML model.

- Table II presents nice summary on the performance of the model. As expected for most of ML potentials, mechanical properties (also Fig. S3) are accurately described by the ML models (GAP and ACE) compared to DFT reference data. Nevertheless, we can see that ACE model consistently underestimate energy related predictions, contrasting to the high accuracy of the GAP model. Most likely, this is related to limitation on the learning capacity of the chosen model. As highlighted by the authors, here, the only comparison that should be considered is ACE to DFT(SCAN), while the apparent agreement of ACE with experiment is due to serendipity.

-The authors stated "How- ever, for properties such as the bulk modulus and surface energies of diamond, the agreement with experiment is better than that of GAP-18." This should be removed given that is a misleading comment. The apparent agreement of ACE results with experiment is due to serendipity. There is no fundamental reason why ACE predictions are comparable with experiment. IF ACE was generating predictive simulations, it should recover SCAN predictions.

Response to Reviewers' Comments – Manuscript NCOMMS-23-30101

We thank all three reviewers for their helpful and constructive feedback. In the following, we quote the reviewers' reports in full, and our responses are interspersed in **blue**. Action taken is indicated in **red**.

Reviewer #1 (Remarks to the Author):

The authors report the state of the art of machine learning technique applied to a series of Si–O system. Although underlying technique of the paper is already reported in ref 31, I positively recommend the ms. for publication in Nature Communications with minor revisions because the introduction of large-scale active learning is quite important and the application to a wide range of Si–O system including amorphous silicon monoxide (please see ref. 15; Hirata, A. et al. Atomic-scale disproportionation in amorphous silicon monoxide. Nat. Commun. 7, 11591 (2016)) is very impressive. I believe that the contents are suitable for a wide range of broad readers of Nature Communications because Si–O system is very important as the authors mentioned in the introduction part of the ms.

Response: We thank the reviewer for their positive recommendation, and for emphasising the importance of the Si–O system studied in this work.

1. I recommend the authors to refer the following article because this article reported the landmark study on the structure of amorphous silicon monoxide, although the content is controversial. Greaves, G. N. EXAFS and the structure of glass. J. Non-Cryst. Solids 71, 203–217 (1985).

Response: We thank the reviewer for pointing out this reference.

Action taken: We now cite the mentioned reference in the Introduction (new ref. 13, p. 1).

2. I strongly recommend the authors to calculate neutron structure factor for amorphous silica because neutrons are sensitive to oxygen atoms. This is helpful supplemental data for a benchmark. Numerical data is available at the following web site.
<http://www.alexhannon.co.uk/DBindex.htm>

Response: This is a good idea, and indeed provides an additional way of validating the potential.

Action taken: We added Supplementary Fig. 14 (p. S32), which contains neutron and x-ray structure factors for amorphous structures generated by the ACE model, as well as by the hybrid protocol described in Ref. 31. We added the following comment to the Discussion section:

“The potential also underestimates the FSDP height of amorphous SiO₂ (Supplementary Fig. 14), as already observed for our earlier GAP model.³¹ Further research is required to identify the origin of this underestimation.” (p. 9)

3. The authors may be interested in the following article. Onodera, Y. et al., Understanding diffraction patterns of glassy, liquid and amorphous materials via persistent homology analyses. *J. Ceram. Soc. Jpn.* 127, 853–863 (2019).

Response: We thank the reviewer for sharing this article. The methodology is highly interesting and we hope that we can use this in future with our structure models to achieve a better understanding of the structure of amorphous silica.

Reviewer #2 (Remarks to the Author):

The authors present the development and application of a machine-learned interatomic potential for Si-O systems. The results are scientifically sound and the paper is written in a compelling and thorough way that makes it easy to follow and enjoyable to read. While I am not 100% sure that the novelty of the work compared to previous works justifies the publication in such a high-impact journal, the importance of the material system combined with the timeliness of the approach does. For this reason, I recommend publishing this paper.

Response: Thank you very much.

Comments:

- To claim that the potential is suitable to study surfaces, I think it is critical to show that surface reconstructions and relaxations are properly captured. Fig. 4a only shows DFT single-point results for the unrelaxed and relaxed ACE structures. How far off are the relaxed DFT values? Moreover, does the potential capture reconstructions such as six-membered rings for the α -quartz (001) surface [<https://doi.org/10.1039/B701176H>] or the 2x1 reconstruction for (001) Si surfaces?

Response: Surface relaxations are indeed an important point, which we realise we did not cover in sufficient detail in the initial submission. Regarding the first point, in Fig. 4a, the relaxation of the α -quartz surfaces by DFT leads to only minor differences in the surface energy, viz. $0.14 \text{ eV}/\text{\AA}^2$ for (001) and at most $0.04 \text{ eV}/\text{\AA}^2$ for all other surfaces considered. We did not observe significant structural changes (reconstructions etc.) during DFT relaxation. Regarding the second point, in both cases the surface reconstructions mentioned by the reviewer are correctly predicted as being energetically more favourable than the cleaved surfaces.

Furthermore, given that a strong focus of our potential is on amorphous silica surfaces, we also looked at surface reconstructions of these. For this purpose, we cut out a sphere from a bulk amorphous structure, which introduces a highly unfavourable surface with many wrongly coordinated atoms. We have equilibrated this surface and show in Supplementary Fig. 3 (p. S22) that the number of wrongly coordinated atoms is strongly decreasing over the simulation time. This also indicates a promising behaviour for surface reconstructions of arbitrary amorphous surfaces.

Action taken: We added Supplementary Fig. 3 (p. S22), which shows the reconstructions as well as the surface energies for the (001) α -quartz and the (100) silicon surface, and we updated Fig. 4a in the main text (p. 6). Moreover, we added three comments to the results part:

“The ACE model is also able to predict the stability of the reconstructed α -quartz (001) surface (Supplementary Fig. 3a–b).” (p. 5)

and

“In Supplementary Fig. 3e–f, we demonstrate the effectiveness of the ACE potential

in reconstructing artificial amorphous surfaces. To this end, we cut out a spherical cavity from an amorphous bulk structure. Upon heating, the amount of incorrectly arranged atoms near the surface ($< 3 \text{ \AA}$) decreases over time, indicating that the surface atoms are undergoing significant reconstructions.” (p. 6)

as well as

“Moreover, the potential captures the reconstruction of the diamond-type silicon (100) surface (Supplementary Fig. 3c–d).” (p. 6)

- Some simple tests in the spirit of Fig. 4a for SiO₂/Si interface energies would strongly support that the segregation of SiO₂ and Si is properly captured by the potential.

Response: This is a good point. We constructed interfaces between amorphous and crystalline silica and silicon by merging models of these structures together. We calculated interface energies with DFT and the ACE potential. The interface energies predicted by the ACE potential agree well with DFT, especially given the wide range of interface energies covered in our tests.

Action taken: We added Supplementary Fig. 10 (p. S28), which shows the interface energies for a wide range of silica–silicon interfaces. We show these results both for relaxed and unrelaxed interfaces, and for DFT as well as for ACE.

- A stunning result obtained with the GAP-18 for silicon is that it is able to describe the polyamorphic transition from LDA-Si to HDA-Si – even though no high-pressure amorphous structures were included in the fit data base [<https://doi.org/10.1038/s41586-020-03072-z>]. Moreover, the GAP-18 potential correctly describes E(V)-curves for the high-pressure st12 and bc8 phases [<https://doi.org/10.1103/PhysRevX.8.041048>], in contrast to the ACE potential. Since the GAP-18 data base is part of the data base for the present potential, but the major shortcoming of the present potential is that high-pressure phases of pure Silicon are poorly described, I wonder where this loss of transferability comes from. Is this a consequence of the GAP vs ACE framework or does the other fit data impede the transferability? Does the current potential capture the polyamorphic transition? In my opinion, a more through discussion of this aspect might help other researchers in the quest of designing truly general-purpose potentials.

Response: Regarding the first question about where the loss of transferability comes from, we believe that it is caused by the additional fitting data. The ACE framework is unlikely to be the reason, as it has already been successfully used to fit the GAP-18 database [Lysogorskiy et al., *npj Comput. Mater.*, 7, 97 (2021)] with similar accuracy as the GAP. The lower accuracy in the silicon part is rather due to the extension to two elements and the focus on SiO₂ in the overall database.

Regarding the polyamorphic transition, yes the potential is able to describe the transition from low density to high density amorphous silicon. However, it does not show the crystallisation around 20 GPa observed for GAP-18. We would like to point out that this is not the purpose of the potential – there are good potentials available for pure silicon, and for a study of only the elemental system those are likely to be the preferred option at this point. Of course, in the future, it would be very interesting to extend the current potential

to describe high-pressure silicon, arriving at the “truly general-purpose” type of potential pointed out by the reviewer.

Action taken: We revised the discussion regarding the accuracy for silicon and extended it (in line with the referee’s comments):

“This is not a principal shortcoming of ACE compared to GAP: it was already shown that it is possible to fit an ACE potential with similar numerical accuracy as GAP-18 to the same training database.³⁵ Instead, the lower accuracy might be caused by the extension of the database to a second element, compared to the GAP-18 one, and its strong focus on the SiO₂ part of the configurational space. We assume that this causes, in turn, a less accurate description of the configurational space of the elemental species. Larger training databases might help to overcome this issue in future.” (p. 7)

We added Supplementary Figure 7 (p. S25) containing an illustration of the polyamorphic transition with the ACE and analysis regarding coordination numbers as well as atomic volume as dependent on the pressure. We also added a short sentence regarding this in the main part:

“ [...], and upon compressing α -Si we do not observe the eventual crystallisation that is described by Si-GAP-18 (Supplementary Fig. 7).⁶²” (p. 7)

- How do the results of Fig. 5 depend on the system size?

Response: The results of Fig. 5 are nearly independent of the system size. Only for the ratio of the interface area between Si and SiO₂ we are able to observe small differences with different system sizes. However, for all other properties, differences are negligible.

Action taken: We added Supplementary Fig. 8 (p. S26) to the manuscript, which shows the structure factors, interface-to-volume ratios and energies of formation for system sizes of 43,904, 221,184, 432,000, and 877,952 atoms, respectively.

- Ref. 3 in the SM “ $\{\mathrm{SiO}\}_2$ ” \rightarrow SiO₂

Response: We thank the reviewer for pointing out this typo.

Action taken: We corrected the formatting on p. S38.

- The “construct surface mesh” modifier results in OVITO should depend on the probe sphere radius. For the sake of reproducibility, I recommend adding this parameter to the methods section.

Response: We added the specifications for the ConstructSurfaceMesh modifier in the Methods section. We changed the procedure slightly compared to the initial submission to obtain an improvement in the calculation of the interface area. This improvement is only minor and the overall conclusions are similar.

Action taken: We extended the Methods section (p. 10) and updated Fig. 5c (p. 7).

Reviewer #3 (Remarks to the Author):

The authors present an ACE Si-O model trained using active learning on DFT(SCAN) reference data. One of the prominent contributions is that the reported ML potential is two orders of magnitude faster than GAP model at the expense of a small reduction in accuracy. The article presents a thorough analysis of the materials' properties using such model. In general, it provides good results, making it of interest for the materials' science users. Nevertheless, this work only focuses on the analysis of the models' characterization and not on a specific problem that, for example, reveals a new understanding on the dynamics of Si-O material.

Response: We thank the reviewer for their positive comments. We also appreciate the fact that they are challenging us to provide clearer insights into a specific application problem, which we have now addressed during revision.

Specifically, we have now carried out new analyses of the dynamical crystallisation process in SiO structures. We achieved this by heating the SiO structures above the melting point of silicon and then quenched it rapidly by 200 K within 20 ns. During the quench, the silicon-rich regions partially crystallised. This process is characterised in the newly added Fig. 6. For example, we built a simple thermodynamic model to understand the conditions under which the amorphous Si crystallises into the SiO structures.

Action taken: We added a new figure (Fig. 6, p. 9) along with a paragraph providing information about it. Additionally, Supplementary Figures 12 (p. S30) and 13 (p. S31) have been added, offering details about the process of crystallisation and interface energies. Supplementary Note 2 contains a description of our thermodynamic model for the Si crystallisation.

Some further comments are presented below:

-When citing the supplementary material, please state the specific section.

Action taken: We added the relevant specific references.

-Table I is a good overview of the accuracy, but what is more informative to assess accuracy are the forces, this because MD simulations only used forces to sample the phase space. Hence, I would recommend to also include force errors in this table.

Response: We agree that the forces are crucial for MD simulations. Whilst we had previously shown them in the Supplementary Material (Supplementary Tab. 4), we now include them in the main part following the referee's suggestion.

Action taken: We extended Table 1 by including force errors (p. 4).

-Fig. 2 Phase diagram: Here, we can see that qualitatively, the ACE model can recover the phase diagram, nevertheless, it shows a severe deficiency predicting the transition liquid \leftrightarrow solid of $\sim 400\text{K}$ in the 0-1GPa region. This is a common issue in ML potentials trained on undersampled liquid phases. Meaning, the ML mostly learn the solid phases. The same happens with water models, where the simulations tend to generate icy phases

instead of the proper liquid state.

Response: We believe that the melting-point underestimation is likely caused by the underlying DFT exchange-correlation functional. We note that we calculated the free energy of the liquid and crystalline phases separately using thermodynamic integration, *i.e.*, without a direct simulation of the transition of the two phases, thereby ruling out dynamic effects. However, undersampling may indeed result in an imprecise description of the liquid phase.

To estimate the error of the ACE model for the liquid phase, one can compute energy errors for snapshots from MD simulations which are part of the ACE-MD test set for amorphous structures (*cf.* Supplementary Table 4). The energy RMSE for the 3,000 K structures is approximately 3.7 meV/atom. Assuming that the energy of the liquid phase is overestimated by ACE, correcting it by 3.7 meV/atom would produce the phase diagram depicted in Supplementary Fig. 19 (p. S36). However, the melting point of 2,260 K thus obtained still deviates from experiment by over 260 K – which, in our view, can be attributed to the underlying exchange-correlation functional data.

Action taken: We added Supplementary Fig. 19 (p. S36) as well as the following paragraph to Supplementary Notes 2:

“Another point is that the melting point is significantly overestimated by the ML potential at low pressures. This can, in principle, be due to two reasons, namely the exchange–correlation functional and an undersampling of the liquid phase space. The latter would likely correspond to a high error in the prediction of the energy of the liquid phase. To assess this, we analyse the 3000 K ACE-MD test set (see Supplementary Table III) for the liquid phase, which has an energy RMSE error of 3.7 meV/atom. When we assume that the energy of the liquid phase is overestimated by this value by the potential, and correct the phase diagram accordingly, we obtain the phase diagram shown in Supplementary Fig. 19. In this phase diagram, the melting point is still overestimated by 260 K, which thus indicates that (at least partially) the exchange-correlation functional is likely the reason for the deviation of the predicted melting point from experiment.” (p. S12)

-The article presents an interesting manual-fine tuning of the model via manipulating the different penalties on the loss function for the different phase states. This corrects some of the boundaries in the phase diagram. This could be a valid non-machine-learned way to compensate undersampled phases. Although, a full analysis on the rest of physical properties is required to assess if the model continues to be physically valid.

Response: This is a very interesting point indeed. We would like to point out, for full clarity, that our Supplementary Fig. 1 analyses the uncertainty of the phase diagram by shifting the free energies of certain phases to higher values as a subsequent correction. The interatomic potential is not directly modified in this study. Nevertheless, in future work, it would be interesting to fine-tune the potential itself to see whether that might improve agreement with experimental results.

Action taken: We revised the caption of Supplementary Fig. 1 (p. S20) to make clear that the potential model itself remains unchanged, and that we added the penalty to the free

energies predicted using the model.

-The authors stated: “We thus conclude that the quantitative deviation seen in Fig. 2 is due to the inaccuracy of the underlying exchange–correlation functional, rather than indicating a shortcoming of the ACE approach.” In general, a more elaborated note should be added to discuss the accuracy of the reference DFT data, this because it is well known that, even though the SCAN functional provides accurate energies, its gradients (i.e. forces) are problematic and highly depend on the stability of the implementation. Thereby, this could be a potential source of predictive error beyond the ML model.

Response: We thank the reviewer for pointing this out. Indeed, even though the SCAN functional provides accurate energies and significantly outperformed other functionals in our tests for silica (and therefore was our DFT approach of choice), there might still be remaining errors for the forces.

Action taken: We added the sentence

”This is an example of the more general problem that any issues with the ground-truth method (e.g., numerical instabilities) will translate into the ML model.”

on p. 3 to extend the discussion regarding possible problems with underlying data.

- Table II presents nice summary on the performance of the model. As expected for most of ML potentials, mechanical properties (also Fig. S3) are accurately described by the ML models (GAP and ACE) compared to DFT reference data. Nevertheless, we can see that ACE model consistently underestimate energy related predictions, contrasting to the high accuracy of the GAP model. Most likely, this is related to limitation on the learning capacity of the chosen model. As highlighted by the authors, here, the only comparison that should be considered is ACE to DFT(SCAN), while the apparent agreement of ACE with experiment is due to serendipity. The authors stated “However, for properties such as the bulk modulus and surface energies of diamond, the agreement with experiment is better than that of GAP-18.” This should be removed given that is a misleading comment. The apparent agreement of ACE results with experiment is due to serendipity. There is no fundamental reason why ACE predictions are comparable with experiment. IF ACE was generating predictive simulations, it should recover SCAN predictions.

Response: We agree that it seems like the learning capacity of the model reached a limit, given the broad part of configurational space which is covered by the database. We acknowledge that the comment regarding the agreement with experiment may have been inadvertently misleading, and we revised the manuscript accordingly to ensure a fair comparison to GAP-18.

Action taken: We extended the discussion of shortcomings of the potential for silicon and possible reasons for this (p. 7). We removed the statement mentioned by the referee.

In conclusion, we thank all three referees again for their constructive comments and for their helpful suggestions. We are convinced that the manuscript has improved further based on their feedback.

REVIEWERS' COMMENTS

Reviewer #1 (Remarks to the Author):

The authors have answered all my questions, and made the corresponding changes. Now I recommend the revised manuscript for publication in Nature Communications.

Reviewer #2 (Remarks to the Author):

The authors addressed all the points I raised very thoroughly. I recommend the publication in its present form. Excellent work!